# Learning Representations by Contrasting Clusters While Bootstrapping Instances

## Abstract

Learning visual representations using large-scale unlabelled images is a holy grail for most of computer vision tasks. Recent contrastive learning methods have focused on encouraging the learned visual representations to be *linearly separable* among the individual items regardless of their semantic similarity; however, it could lead to a sub-optimal solution if a given downstream task is related to non-discriminative ones such as cluster analysis and information retrieval. In this work, we propose an advanced approach to consider the instance semantics in an unsupervised environment by both i) *Contrasting* batch-wise **C**luster assignment features and ii) *Bootstrapping* an **IN**stance representations without considering negatives simultaneously, referred to as C2BIN. Specifically, instances in a mini-batch are appropriately assigned to distinct clusters, each of which aims to capture apparent similarity among instances. Moreover, we introduce a multi-scale clustering technique, showing positive effects on the representations by capturing multi-scale semantics. Empirically, our method achieves comparable or better performance than both representation learning and clustering baselines on various benchmark datasets: CIFAR-10, CIFAR-100, and STL-10.

## 1 Introduction

Learning to extract generalized representations from a high-dimensional image is essential in solving various down-stream tasks in computer vision. Though a supervised learning framework has shown to be useful in learning discriminative representations for pre-training the model, expensive labeling cost makes it practically infeasible in a large-scale dataset. Moreover, relying on the human-annotated labels tends to cause several issues such as class imbalance (Cui et al., 2019), noisy labels (Lee et al., 2019), and biased datasets (Bahng et al., 2019). To address these issues, self-supervised visual representation learning, which does not require any given labels, has emerged as an alternative training framework, being actively studied to find a proper training objective.

Recently, self-supervised approaches with contrastive learning (Wu et al., 2018; Chen et al., 2020a; He et al., 2020) have rapidly narrowed the performance gap with supervised pre-training in various vision tasks. The contrastive method aims to learn *invariant mapping* (Hadsell et al., 2006) and *instance discrimination*. Intuitively, two augmented views of the same instance are mapped to the same latent space while different instances are pushed away. However, aforementioned instance discrimination does not consider the semantic similarities of the representations (e.g., same class), even pushing away the relevant instances. This affects the learned representations to exhibit uniformly distributed characteristics, proven by the previous works (Wang & Isola, 2020; Chen & Li, 2020).

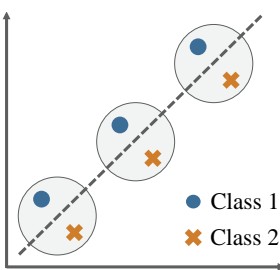

Figure 1: Though illustrated 2D representations are linearly separable, irrelevant instances are clustered together.

We point out that this *uniformly distributed* characteristic over instances can be a fundamental limitation against improving the learned representation quality. For instance, consider the representations illustrated in Fig. 1. It indicates a simple case where linearly separable representations do not always guarantee that they can

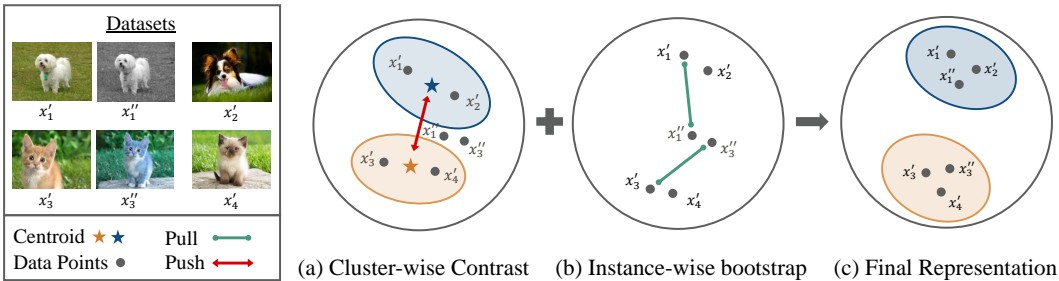

Figure 2: Visual illustration of how our method leads to both linearly separable and clusterable representations. While semantically unrelated samples are pushed apart with the cluster-wise contrastive loss, the invariant mapping can be maintained by our instance-wise bootstrapping loss.

be properly clustered, which is not appropriate for non-discriminative downstream tasks such as information retrieval, density estimation, and cluster analysis (Wu et al., 2013). In response, we start this work by asking: How can we learn the representations to be properly clustered even without the class labels?

In this work, we propose a self-supervised training framework that makes the learned representations not only linearly separable but also properly clustered, as illustrated in Fig. 2. To mitigate the uniformly distributed constraint while preserving the invariant mapping, we replace the instance discrimination with an instance alignment problem, pulling the augmented views from the same instance without pushing away the views from the different images. However, learning the invariant mapping without discrimination can easily fall into a trivial solution that maps all the individual instances to a single point. To alleviate this shortcoming, we adopt a bootstrapping strategy from Grill et al. (2020), utilizing the Siamese network, and a momentum update strategy (He et al., 2020).

In parallel, to properly cluster the semantically related instances, we are motivated to design additional cluster branch. This branch aims to group the relevant representations by softly assigning the instances to each cluster. Since each of cluster assignments needs to be discriminative, we employ the contrastive loss to the assigned probability distribution over the clusters with a simple entropy-based regularization. In the meantime, we constructed the cluster branch in multi-scale clustering strategy where each head deals with a different number of clusters (Lin et al., 2017). Since there exists a various granularity of semantic information in images, it helps the model to effectively capture the diverse level of semantics as analyzed in Section 4.5.

In summary, our contributions are threefold, as follows:

- We propose a novel self-supervised framework which contrasts the clusters while bootstrapping the instances that can attain both linearly separable and clusterable representations.

- We present a novel cluster branch with multi-scale strategy which effectively captures the different levels of semantics in images.

- Our method empirically achieves state-of-the-art results in CIFAR-10, CIFAR-100, and STL-10 on representation learning benchmarks, for both classification and clustering tasks.

## 2 RELATED WORK

Our work is closely related to unsupervised visual representation learning and unsupervised image clustering literature. Although both have a slightly different viewpoints of the problem, they are essentially similar in terms of its goal to find good representations in unlabelled datasets.

Instance-level discrimination utilizes an image index as supervision because it is an unique signal in the unsupervised environment. NPID (Wu et al., 2018) firstly attempts to convert the class-wise classification into the extreme of instance-wise discrimination by using external memory banks. MoCo (He et al., 2020) replaces the memory bank by introducing a momentum encoder that memorizes knowledge learned from the previous mini-batch. SimCLR (Chen et al., 2020a) presents

that it is crucial for representation quality to combine data augmentations using a pretext head after the encoder. Although recent studies show promising results on benchmark datasets, the instance-wise contrastive learning approach has a critical limitation that it pushes away representations from different images even if the images have similar semantics, e.g., belonging to the same class.

Cluster-level bootstrapping is an alternative paradigm that enhancing the initial bias of the networks can be useful in obtaining a discriminative power in visual representations, since convolutional neural networks work well on capturing the local patterns (Caron et al., 2018). In the case of using pseudo-labels, K-means (Caron et al., 2018) or optimal transport (Asano et al., 2019; Caron et al., 2020) are commonly adopted for clustering. On the other hand, soft clustering methods have also been actively studied to allow flexible cluster boundaries (Ji et al., 2019; Huang et al., 2020). Recently, a 2-stage training paradigm has been proposed to construct the cluster structure initialized from the representations learned by instance discrimination (Gansbeke et al., 2020).

## 3 METHOD

Our work is motivated by an observation from SupCLR (Khosla et al., 2020), which additionally pulls the representations together from different instances by using groundtruth labels. However, directly applying this idea in an unsupervised environment with pseudo-labels is challenging, because small false-positive errors at the initial step can be gradually spread out, degrading the quality of final representations.

Instead, the main idea of our approach avoid pushing away those instances close enough to each other. To validate this idea, we conducted a toy experiment that a pulling force is only corresponding to two augmented views of the same image while not pushing the images within the same class by using the groundtruth label. We found that its classification accuracy increases over 5% on STL-10 datasets compared to that of SimCLR (Chen et al., 2020a). Inspired by this experiment, we design our model (i) not to push away relevant instances with our instance-alignment loss (Section 3.2) while (ii) discriminating the representations in a cluster-wise manner. (Section 3.3-3.4).

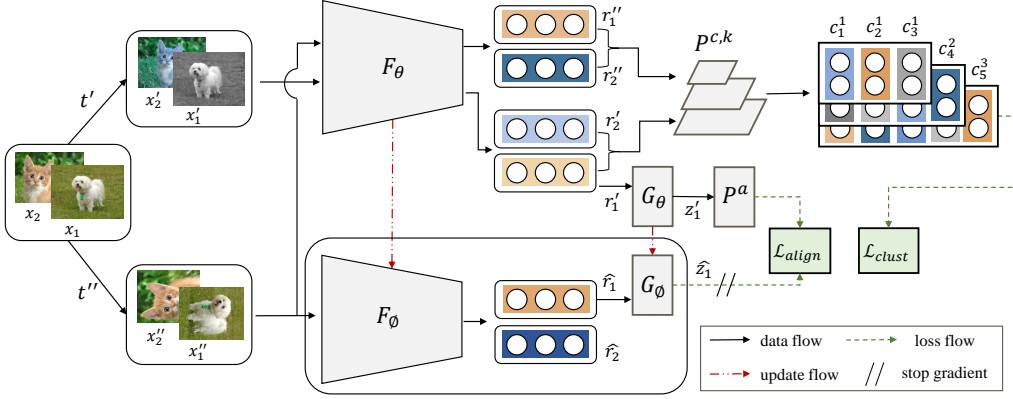

Figure 3: Overall architecture of our proposed C2BIN.

### 3.1 PRELIMINARIES

As shown in Fig. 3, we adopt stochastic data augmentation algorithms (Chen et al., 2020a; He et al., 2020; Chen et al., 2020b; Caron et al., 2020) to generate two different augmented views $x_i'$ and $x_i''$ of the same image $x_i \sim \mathcal{X} = \{x_1, x_2, ..., x_N\}$ where $N$ is the number of *unlabelled* images. Inspired by Luo et al. (2018); Grill et al. (2020), C2BIN consists of an instance predictor $P^a(\cdot)$, cluster predictors $P^{c,k}(\cdot)$, and two Siamese networks called the runner $E_\theta(\cdot)$ and the follower $E_\phi(\cdot)$, respectively. The runner $E_\theta$ is rapidly updated to find the optimal parameters $\theta^*$ over the search spaces, while the follower $E_\phi$ generates the target representations for the $E_\theta$. The $E_\theta$ is composed of the two neural functions: encoder $F_\theta(\cdot)$ and instance projector $G_\theta^a(\cdot)$, and vice versa for the follower

$E_\phi$. To bootstrap the instance-level alignment, $E_\theta$, $E_\phi$, and $P^a$ are used. Afterwards, $F_\theta$ and $P^{c,k}$ are utilized to contrast the cluster-wise features.

## 3.2 BOOTSTRAPPING LOSS OF INSTANCE REPRESENTATIONS

Given an image $x \sim \mathcal{X}$, we can obtain two augmented views $x' = t'(x)$ and $x'' = t''(x)$ where $t'$ and $t''$ are sampled from a set of stochastic data augmentations $\mathcal{T}$ as mentioned above. Even though augmented views are distorted, they should contain similar semantics, and the learned representations should be closely aligned in the latent space. For training, we forward $x''$ through the follower $E_\phi$ to obtain target representations at an instance level; the runner $E_\theta$ aims to make the embedding vector of $x'$ closer to them. That is, we first extract image representations $r = F_\theta(x') \in \mathbb{R}^{d_r}$ where $d_r$ is the number of dimensions of our representations. Afterwards, we introduce a pretext-specific instance-wise projector $G_\theta^a(\cdot)$ and then obtain pretext embedding vectors $z_a = G_\theta^a(r) \in \mathbb{R}^{1 \times d_a}$; the target pretext vectors $\hat{z}^a$ can be obtained using the same procedure by $E_\phi$. Motivated from Grill et al. (2020), we calculate our alignment loss as the cosine distance as

$$\mathcal{L}_{align} = 1 - \frac{P^a(z_a) \cdot \hat{z}_a}{||P^a(z_a)||_2 ||\hat{z}_a||_2}, \tag{1}$$

where $P^a(z_a), \hat{z}_a \in \mathbb{R}^{1 \times d_a}$ and we adopt the number of dimensions of projected features $d_a$ as in Chen et al. (2020a;c).

## 3.3 CONTRASTIVE LOSS OF BATCH-WISE CLUSTER ASSIGNMENTS

Our high-level motivation of this branch is that an image feature $r$ can be represented as the combination of cluster features capturing local patterns. However, grouping similar images conflict with the instance-level invariant mapping; therefore, we introduce an additional branch which contains cluster predictor $P^{c,k}(\cdot)$ after the encoder $F_\theta(\cdot)$. The cluster predictor $P^{c,k}$ is a linear function whose takes $r_i$ as an input and transform it to a $K$-dimensional output vector. Therefore, $z_i^c = P^{c,k}(r_i)$ represents a degree of confidence for the $i$-th image representations $r_i$ to belong to the $k$-th cluster feature, i.e.,

$$z_i^c = [z_{i,1}^c, z_{i,2}^c, ..., z_{i,k}^c] \in \mathbb{R}^{1 \times K}, \tag{2}$$

where $z_i^c$ indicate a cluster membership distribution of the given image $x_i$. Since we sample $n$ items for training, $Z^c \in \mathbb{R}^{n \times K}$ is the set of memberships distribution of the given mini-batch. Now we define batch-wise cluster assignment vectors (BCAs) $c_k$ as

$$c_k = Z_{:,k}^c = \begin{bmatrix} z_{1,k}^c \\ \vdots \\ z_{n,k}^c \end{bmatrix} \in \mathbb{R}^{n \times 1}, \tag{3}$$

which indicates how much the $k$-th cluster is mapped by images in the mini-batch. Although $c_k$ will dynamically change as a new mini-batch is given, the same cluster features between differently augmented views from the same image should be similar while pushing away the others to capture diverse patterns. To this end, we simply utilize the contrastive loss between the BCAs as

$$\mathcal{L}_{clust}^{bca} = \frac{1}{K} \sum_{i=1}^{K} -\log \left( \frac{\exp(c_i' \cdot c_i''/\tau)}{\sum_{j=1}^{K} \mathbb{1}_{[j \neq i]} \exp(c_i' \cdot c_j''/\tau)} \right), \tag{4}$$

where $\tau$ indicates a temperature value. The vectors $c'$ and $c''$ are outputs of $P^{c,k}$ following the encoder $F_\theta$ by taking $x'$ and $x''$ respectively.

Unfortunately, most of the clustering-based methods suffers from falling into a degenerate solution where the majority of items are allocated in a few clusters, especially in an unsupervised environment. To mitigate this issue, we first compute the mass of assignment to $k$-th cluster as $s_k = \sum_i^N c_k(i)$ where $c_k(i)$, indicating each element of $c_k$. Afterwards, we encourage $r_i$ to be stochastically activated for diverse cluster features as much as possible by maximizing an entropy of $s$. To this end, we formulate the cluster loss function as

$$\mathcal{L}_{clust} = \mathcal{L}_{clust}^{bca} - \lambda_{ent} H(s), \tag{5}$$

where $H$ indicates an entropy function as $H(s) = -\sum_i^K s_i \log s_i$ and $\lambda_{ent}$ is the weight value for the regularization term.

### 3.4 Multi-scale Clustering Strategy

The multi-scale clustering strategy has often been used in prior research (Vaswani et al., 2017; Asano et al., 2019), leveraging the ensembling effect. Extending this strategy, we propose a multi-scale clustering strategy for our task. Although contrasting between the BCAs encourages our model to capture various aspects of local patterns, the performance may be sensitive to the number of clusters $k$. To address this issue, we introduce a set of the cluster branches that have a different number of cluster assignments in each scale. To this end, we reformulate $\mathcal{L}_{clust}$ as

$$\mathcal{L}_{clust} = \sum_k \mathcal{L}_{clust}^k, k \in K. \tag{6}$$

In this work, we use various values of $k$, e.g., $K = \{32, 64, 128\}$.

### 3.5 Total Objective

Finally, our total objective function is written as

$$\mathcal{L}_{total} = \mathcal{L}_{align} + \lambda_{clust}\mathcal{L}_{clust}. \tag{7}$$

The parameters of the follower $E_\phi$ gradually reflect those of the runner via

$$\phi \leftarrow \gamma\phi + (1 - \gamma)\theta, \tag{8}$$

where $\gamma$ indicates a momentum factor.

## 4 Experiments

This section presents the experimental evaluation of C2BIN on the standard benchmark datasets including CIFAR-10, CIFAR-100, STL-10, and ImageNet, which are commonly adopted in both self-supervised representation learning and unsupervised image clustering literature.

In Sections 4.1-4.3, we compare C2BIN with several representation learning methods and unsupervised clustering methods to verify that our model can yield both linearly separable and clusterable representations. Afterwards, Section 4.4 studies the robustness of C2BIN in an class-imbalanced setting. Lastly, Section 4.5 presents an ablation study for in-depth analysis of our model behaviour.

### 4.1 Representation Learning Tasks on Unified Setup

**Experimental setup**. Because previous studies have their own experimental settings in terms of datasets and backbone architectures, we prepared for a unified experimental setup for the fair comparison , as follows. We first employ the ResNet-18 architecture as the backbone architecture, following Wu et al. (2018). We used 3 standard benchmark datasets; CIFAR-10, CIFAR-100, and STL-10 for this experiment. All baselines are trained by using the identical data augmentation techniques, as used in Chen et al. (2020a;c). Further training details can be found in Appendix A.1.

**Evaluation metrics**. We adopt three standard evaluation metrics: linear evaluation protocol (**LP**) (Zhang et al., 2017), k-nearest-neighbour (**kNN**) classifier with $k = 5$ and $k = 200$. For the LP, we follow the recipe of Grill et al. (2020), where we report the best evaluation score over five differently initialized learning rates. For kNN, we follow the settings used in Wu et al. (2018) and the implementation of Asano et al. (2019).

For our baseline methods, we choose SimCLR, MoCo v2, and BYOL, which can work as the state-of-the-art methods for the instance-wise contrastive learning, momentum-based contrastive learning, and instance-wise bootstrapping, respectively. As seen in Tab. 1, C2BIN consistently outperforms all baselines across all benchmark datasets In the case of CIFAR-100, the kNN accuracy of C2BIN significantly improves compared to the baselines while its LP scores consistently increase as well. We conjecture the reason is becuase C2BIN is appropriate to learn a hierarchical structure in a dataset such as CIFAR-100

| Ar. | Method | gap+LP / gap+kNN(k=5) / gap+kNN(k=200) | | |
| --- | --- | --- | --- | --- |
| | | CIFAR-10 | CIFAR-100 | STL-10 |
| ResNet-18 | DC (Caron et al., 2018) | - / - / 67.6 | - | - |
| | NPID (Wu et al., 2018) | - / - / 80.8 | - / - / 51.6 | - |
| | SimCLR (Chen et al., 2020a) | 81.3 / 82.4 / 81.7 | 59.8 / 63.9 / 66.8 | 83.1 / 78.3 / 78.8 |
| | MoCo (He et al., 2020) | 78.1 / 79.7 / 77.7 | 50.2 / 56.7 / 58.2 | 79.4 / 74.3 / 74.2 |
| | MoCo v2 (Chen et al., 2020c) | 81.6 / 81.4 / 83.9 | 61.1 / 62.8 / 63.8 | 83.5 / 78.2 / 78.1 |
| | BYOL † (Grill et al., 2020) | 80.8 / 81.4 / 79.7 | 57.0 / 63.3 / 61.4 | 80.4 / 76.5 / 77.5 |
| | C2BIN [Mean] (Ours) | 81.5 / 84.8 / 84.9 | 61.5 / 72.2 / 72.6 | 83.6 / 79.1 / 80.1 |
| | C2BIN [Best] (Ours) | **82.3 / 85.9 / 85.3** | **62.1 / 73.9 / 73.8** | **84.0 / 79.9 / 80.8** |

Table 1: Comparison with unsupervised representation methods. Note on †: for the fair comparison, we did not used the average gradient trick that was utilized in BYOL (Grill et al., 2020).

## 4.2 REPRESENTATION LEARNING TASKS ON LARGE SCALE BENCHMARK

**Experimental setup**. To compare our method with concurrent and state-of-the-art works in a large scale dataset, we evaluate our method in ImageNet with ResNet-50 architecture as a backbone model. For fair comparison, most of the results are taken from the experiments that models are trained for 200 epochs with a batch size of 256. All of the baselines are trained with identical data augmentation techniques introduced in (Chen et al., 2020a). Further training details can be found in Appendix A.2.

| Method | Epochs | Batch Size | Top-1 accuracy |
| --- | --- | --- | --- |
| NPID (Wu et al., 2018) | 200 | 256 | 56.5 |
| MoCo (He et al., 2020) | 200 | 256 | 60.6 |
| SimCLR (Chen et al., 2020a) | 200 | 256 | 61.9 |
| MoCo v2 (Chen et al., 2020c) | 200 | 256 | 67.5 |
| BYOL † (Grill et al., 2020) | 200 | 256 | 64.3 |
| C2BIN (Ours) | 200 | 256 | 64.4 |
| SimCLR (Chen et al., 2020a) | 400 | 4096 | 68.2 |
| SwAV (Caron et al., 2020) | 400 | 4096 | 70.1 |
| MoCo v2 (Chen et al., 2020c) | 800 | 256 | 71.1 |

Table 2: Linear classifier top-1 accuracy comparison with unsupervised representation methods on ImageNet. Methods are arranged in chronological order. Note on †: BYOL (Grill et al., 2020) does not report the result on ImageNet with the identical experimental setup. Therefore, we adopted the results of BYOL from Zhan et al. (2020), the most widely used open-source library for self-supervised learning, experimented without an average gradient technique for a fair comparison.

Though C2BIN has shown competitive performance compared to the baselines, MoCo v2 (Chen et al., 2020c) outperforms C2BIN on the large-scale dataset, which contradicts the findings in Section 4.1. Since C2BIN utilizes batch-wise clustering techniques to learn the cluster structure, it brings instability to the training process when a large number of cluster size is required compared to the batch size. Still, our method slightly outperforms the state-of-the-art instance bootstrapping method, BYOL (Grill et al., 2020). This result implies that simultaneously learning the cluster structure is not counter-effective to enhance the discriminative power of the instance-wise representation learning method.

## 4.3 IMAGE CLUSTERING TASKS

This section compares our approach to the baselines in the unsupervised image clustering task.

**Experimental setup**. For a fair comparison, we keep most of the implementation details identical to Ji et al. (2019); Huang et al. (2020) except for excluding the use of the Sobel filter. We use the architecture similar to ResNet-34, with the 2-layer MLP for both the instance projector $G_\theta^a(\cdot)$ and

the predictor $P^a(\cdot)$. For the clustering branch, three cluster heads are used as $K = \{10, 40, 160\}$ for CIFAR-10 and STL-10, and $K = \{20, 40, 160\}$ for CIFAR-100. Further training details can be found in Appendix B.1.

**Evaluation metrics**. Three standard clustering performance metrics are used for evaluation: (a) Normalized Mutual Information (**NMI**) measures the normalized mutual dependence between the predicted labels and the ground-truth labels. (b) Accuracy (**ACC**) is measured by assigning dominant class labels to each cluster and take the average precision. (c) Adjusted Rand Index (**ARI**) measures how many samples are assigned properly to different clusters. All the evaluation metrics range between 0 and 1, where the higher score indicates better performance.

| Method | NMI / ACC / ARI | | |
|---|---|---|---|
| | CIFAR-10 | CIFAR-100 | STL-10 |
| K-means | 0.09 / 0.23 / 0.05 | 0.08 / 0.13 / 0.03 | 0.13 / 0.19 / 0.06 |
| DEC  (Xie et al., 2016) | 0.26 / 0.30 / 0.16 | 0.14 / 0.19 / 0.05 | 0.28 / 0.36 / 0.19 |
| DCCM  (Wu et al., 2019) | 0.50 / 0.62 / 0.41 | 0.29 / 0.33 / 0.17 | 0.38 / 0.48 / 0.26 |
| IIC  (Ji et al., 2019) | - / 0.62 / - | - / 0.26 / - | - / 0.61 / - |
| PICA  (Huang et al., 2020) | 0.59 / 0.70 / 0.51 | 0.31 / 0.34 / 0.17 | 0.61 / 0.71 / 0.53 |
| C2BIN [Mean] (Ours) | 0.62 / 0.72 / 0.53 | 0.36 / 0.35 / 0.20 | 0.62 / 0.73 / 0.55 |
| C2BIN [Best] (Ours) | **0.63 / 0.73 / 0.55** | **0.38 / 0.38 / 0.22** | **0.64 / 0.75 / 0.57** |

Table 3: Comparison with end-to-end unsupervised representation methods in the clustering benchmark. The results of previous methods are taken from Huang et al. (2020). We append full comparison results in Appendix (Table 11).

As shown in Table 3, C2BIN outperforms the state-of-the-art clustering performance in all datasets by a significant margin, showing its capability of grouping the semantically related instances to distinct clusters. Moreover, C2BIN is shown to be robust, given that the averaged performance over five random trials even surpasses the best results from the previous literature.

## 4.4 CLASS-IMBALANCED EXPERIMENTS

Unlike the standard benchmark datasets we used, it is often the case that the real-world image dataset is severely imbalanced in terms of its underlying class distribution. Therefore, we conducted additional experiments in a class-imbalanced environment, following the experimental design proposed in Cui et al. (2019).

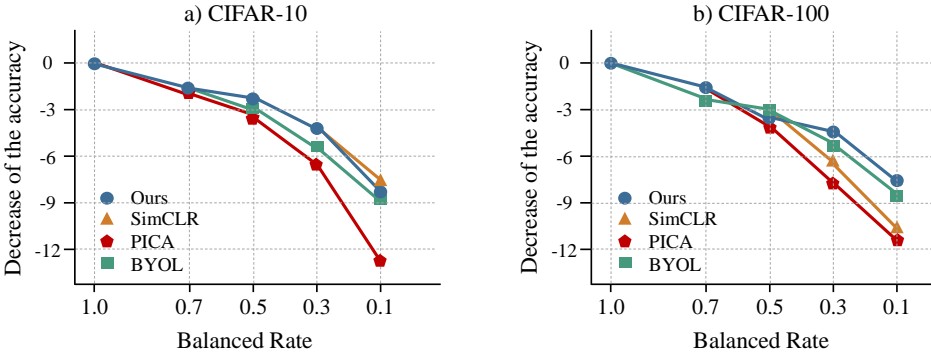

Figure 4: Top-1 accuracy degradation when using the ResNet-18 architecture under a linear evaluation protocol in a class-imbalanced setting.

Fig. 4 demonstrates the classification accuracy degradation in an imbalanced setting. The balanced rate indicates the relative ratio of the largest to the smallest classes. We find that the performance of PICA, the clustering-based method, significantly decreases compared to other baselines as the class imbalance gets apparent. In the case of imbalanced CIFAR-100, SimCLR, which contrasts

all instances within the mini-batch, is shown to get degraded faster than BYOL, which does not consider the relationship between other instances. On the other hand, the accuracy degradation of C2BIN is shown to be minimal for both CIFAR-10 and CIFAR-100, possibly due to our alignment loss (Section 3.2).

## 4.5 DISCUSSIONS

**Qualitative analysis on the learned representations**.

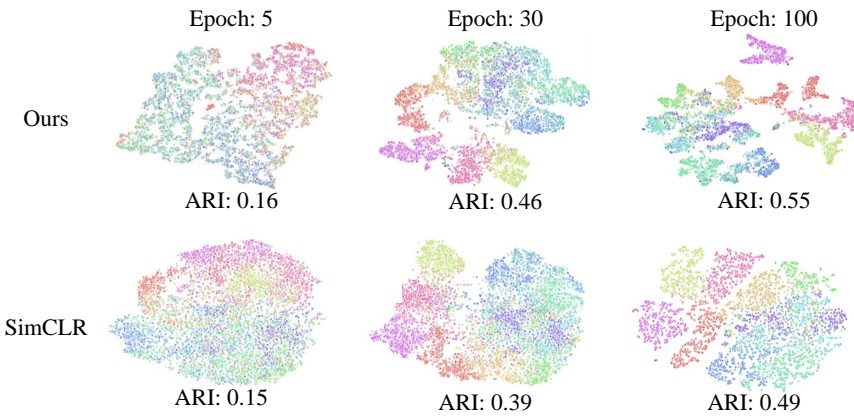

Figure 5: t-SNE 2-D embedding visualization of C2BIN and SimCLR.

For the test data items in STL-10 dataset, we embed their high-dimensional representations obtained by our method and SimCLR in a 2-D space using t-SNE(van der Maaten & Hinton (2008)). As shown in Fig. 5, the representations learned from our model show a clearer cluster structure than SimCLR(Chen et al. (2020a)), as training proceeds.

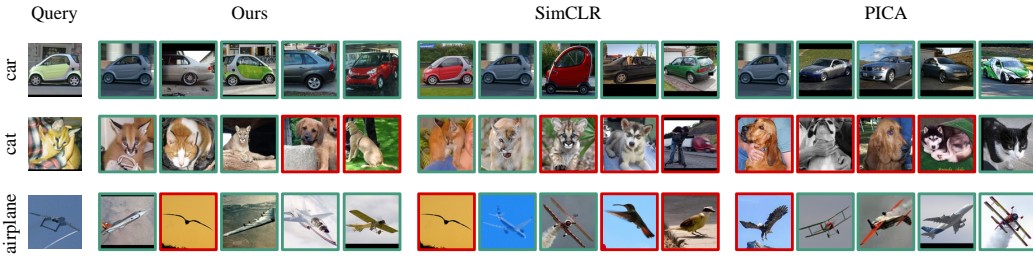

Figure 6: Qualitative comparisons of the top-k retrieved images by C2BIN (Columns 2-6), SimCLR (Columns 7-11), and PICA (Columns 12-17) given a query image (Column 1) from the STL-10 test set where the $k$ is set as $\{1, 2, 10, 50, 100\}$.

To understand the characteristics of both instance-wise alignment and cluster-wise discrimination in a straightforward manner, we conduct the image retrieval experiment. As shown in Fig. 6, our method outperforms two proposed baselines from both perspectives. Since SimCLR only focuses on instance-wise discrimination, it fails to retrieve with a larger value of $k$ with a given query image (e.g., airplane). Likewise, PICA lacks the alignment capability in an instance-wise manner, resulting in poor performance with a lower value of $k$ in contrast to C2BIN. This is also corroborated by the quantitative results in Appendix (Fig. 8).

**Ablation study**. To further verify whether our loss terms are complementary to each other, we perform an ablation study on STL-10 dataset. As we can observe in Table 4(b) and 4(c), a simple integration of the clustering method into the instance-wise bootstrapping (Table 4(a)) can degrade the representation quality unless an appropriate level of granularity is provided. Similar to the results from Asano et al. (2019), using a simple multi-scale clustering branch with a specific number of

clusters (Table 4(d) and (e)) is a more effective strategy than a single-head method. Furthermore, our proposed multi-scale clustering strategy (Table 4(f)) peaks out the best performance since it allows the model to capture the diverse semantic information at a different level. This result justifies our motivation to utilize a clustering strategy in a multi-scale manner.

| | $\mathcal{L}_{align}$ | $\mathcal{L}_{clust}$ | | | | $m_1$ | $m_2$ | $m_3$ |
|---|---|---|---|---|---|---|---|---|
| | Eq. (1) | Eq. (5) | Eq. (6) | multi-scale | K | | | |
| (a) | ✓ | | | | - | 80.4 | 76.5 | 77.5 |
| (b) | ✓ | ✓ | | | $k_1$ | 78.4 (-1.6) | 73.5 (-3.0) | 73.9 (-3.6) |
| (c) | ✓ | ✓ | | | $k_2$ | 81.3 (+0.9) | 76.3 (-0.2) | 77.0 (-0.5) |
| (d) | ✓ | | ✓ | | $k_3$ | 79.3 (-0.9) | 74.4 (-2.1) | 75.5 (-2.0) |
| (e) | ✓ | | ✓ | | $k_4$ | 82.2 (+1.8) | 76.3 (-0.2) | 76.4 (-1.1) |
| (f) | ✓ | | ✓ | ✓ | $k_5$ | **84.0** (+3.6) | **79.9** (+3.4) | **80.8** (+3.3) |

Table 4: Performance improvements due to each of our components. $m_1$, $m_2$, and $m_3$ indicate the linear evaluation protocol (LP), kNN(k=5), and kNN(k=200), respectively. $K$ denotes a set of cluster sizes: $k_1 = \{32\}$, $k_2 = \{128\}$, $k_3 = \{32, 32, 32\}$, $k_4 = \{128, 128, 128\}$, and $k_5 = \{32, 64, 128\}$.

**Visual analysis on multi-scale clustering strategy**. We also show the visual analysis on the multi-scale clustering strategy. Each scale represents the different semantic information as shown in Appendix (Figs. 9, 10, and 11). Combining this semantic difference in each scale prevents our model from binding to a specific number of cluster assignments.

## 5    CONCLUSIONS

In this paper, we proposed a novel approach to improve the existing representation learning with unsupervised image clustering. By integrating the advantages of both literature, we present an advanced self-supervised framework that simultaneously learns cluster features as well as image representations by contrasting clusters while bootstrapping instances. Moreover, in order to capture diverse semantic information, we suggest a multi-scale clustering strategy. We also conduct ablation studies to validate complementary effects of our proposed loss functions.

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

## A  REPRESENTATION LEARNING EXPERIMENTS

### A.1  IMPLEMENTATION DETAILS FOR UNIFIED SETTING

| Hyperparameter | Value |
|---|---|
| Epoch | 500 |
| Optimizer | LARS (You et al., 2017) |
| Learning rate | Constant(0.2) |
| Weight decay | 1e-6 |
| Weight momentum | 0.9 |
| Batch size | 256 |
| Cluster weight ($\mathcal{L}_{clust}$) | 2.0 |
| Entropy weight ($\mathcal{L}_{entropy}$) | 1.0 |
| Architecture | ResNet-18 |
| Representation dim | 512 |
| Instance projector $G_\theta^a$ | MLP(512, 512) ReLU |
| Instance predictor $P_\theta^a$ | MLP(512, 512) ReLU |
| Momentum factor ($\gamma$) | 0.990 |
| Temperature ($\tau$) | 0.5 |
| Cluster head ($K_1$) | 32 (CIFAR-10, CIFAR-100, STL-10) |
| Cluster head ($K_2$) | 64 |
| Cluster head ($K_3$) | 128 |

Table 5: Hyperparameters of backbone models used in the experiment of Section 4.1

| Hyperparameter | Value |
|---|---|
| Epoch | 300 |
| Optimizer | LARS (You et al., 2017) |
| Learning rate | Constant({0.1, 0.2, 0.3, 0.4, 0.5}) |
| Weight decay | 1e-6 |
| Weight momentum | 0.9 |
| Batch size | 256 |
| Architecture | linear without batch-norm layer |

Table 6: Hyperparameters of the linear evaluation protocol used in the experiment of Section 4.1

## A.2 IMPLEMENTATION DETAILS FOR THE LARGE-SCALE SETTING

| Hyperparameter | Value |
|---|---|
| Epoch | 200 |
| Optimizer | SGD |
| Learning rate | 0.03 |
| Learning rate schedule | multiplying 0.1 at 120, 160 epoch. |
| Weight decay | 1e-6 |
| Weight momentum | 0.9 |
| Batch size | 256 |
| Cluster weight ($\mathcal{L}_{clust}$) | 1.0 |
| Entropy weight ($\mathcal{L}_{entropy}$) | 0.05 |
| Architecture | ResNet-50 |
| Representation dim | 2048 |
| Instance projector $G_\theta^a$ | MLP(2048, 4096) ReLU |
| Instance predictor $P_\theta^a$ | MLP(2048, 4096) ReLU |
| Momentum factor ($\gamma$) | 0.990 |
| Temperature ($\tau$) | 0.2 |
| Cluster head ($K_1$) | 512 |
| Cluster head ($K_2$) | 1024 |
| Cluster head ($K_3$) | 2048 |

Table 7: Hyperparameters of backbone models used in the experiment of Section 4.2

| Hyperparameter | Value |
|---|---|
| Epoch | 200 |
| Optimizer | SGD |
| Learning rate | 30.0 |
| Learning rate schedule | multiplying 0.1 at at 60 and 80 epoch |
| Weight decay | 1e-6 |
| Weight momentum | 0.9 |
| Batch size | 256 |
| Architecture | linear without batch-norm layer |

Table 8: Hyperparameters of the linear evaluation protocol used in the experiment of Section 4.2

### A.3 Impact Study for choice of $K$

Although the effectiveness of the multi-scale clustering technique is briefly described in Section 4.5, this section studies performance changes according to the choice of the set $K$.

| | $\mathcal{L}_{align}$ | $\mathcal{L}_{clust}$ | | | | LP (%) |
|---|---|---|---|---|---|---|
| | Eq. (1) | Eq. (5) | Eq. (6) | multi-scale | $K$ | |
| (a) | ✓ | | ✓ | ✓ | $\{32, 64, 128\}$ | 84.0 |
| (b) | ✓ | ✓ | | | $\{8\}$ | 75.28 (-8.72) |
| (c) | ✓ | ✓ | | | $\{16\}$ | 79.06 (-4.94) |
| (d) | ✓ | ✓ | | | $\{512\}$ | 80.01 (-3.9) |
| (e) | ✓ | | ✓ | | $\{8, 8, 8\}$ | 76.08 (-7.92) |
| (f) | ✓ | | ✓ | | $\{16, 16, 16\}$ | 80.08 (-3.92) |
| (g) | ✓ | | ✓ | | $\{512, 512, 512\}$ | 80.50 (-3.5) |
| (h) | ✓ | | ✓ | ✓ | $\{8, 16, 32\}$ | 83.50 (-0.5) |
| (i) | ✓ | | ✓ | ✓ | $\{16, 32, 64\}$ | 83.91 (-0.09) |

Table 9: **An impact study about the choice of $K$ on the STL-10 dataset.** LP indicates an linear evaluation protocol described in Section 4.1.

Table 9 shows how C2BIN's performance is damaged for the linear evaluation protocol (LP) on the STL-10 dataset when the combination of $K$ is changed. In Table 9, (a) is the best score reported in the main paper and can be used as a pivot for comparison. For the rest of them, similar to Section 4.5, we divide experiments into three groups. First, (b)-(d) are matched with the case of attaching single and arbitrary selected cluster size. Unfortunately, this case does not help to improve performances and even dramatically degenerates the representation quality. We guess that attaching a single cluster head after the backbone network makes its representation quality sensitive according to the head size. The second group, (e)-(g), corresponds with the case of multiple but single-scale cluster heads. Although it seems a slight improvement compared to the above-mentioned case, it is difficult to be sufficiently complementary in our setting. We think the effect of the multi-branch clustering seems small because each cluster head can capture similar patterns with others. Lastly, (h)-(i) is mapped to the case of our multiple and multi-scale clustering strategy, showing robust performance in regard to the combination of $K$ if each element of $K$ is assigned in different scales. We guess that the effect of the multi-task learning is maximized because an identical representation vector should be informative enough to satisfy the following clusters, which are from abstracted to detailed.

## B UNSUPERVISED CLUSTERING EXPERIMENTS

### B.1 IMPLEMENTATION DETAILS

| Hyperparameter | Value |
|---|---|
| Epoch | 300 |
| Optimizer | Adam (Kingma & Ba, 2015) |
| Learning rate | Cosine annealing (3e-4, 0) |
| Weight decay | No weight decay |
| Batch size | 256 |
| Cluster weight ($\mathcal{L}_{clust}$) | 1.0 |
| Entropy weight ($\mathcal{L}_{entropy}$) | 1.0 |
| Architecture | ResNet-34 |
| Representation dim | 512 |
| Instance projector $G_\theta^a$ | MLP(512, 512) ReLU |
| Instance predictor $P_\theta^a$ | MLP(512, 512) ReLU |
| Momentum factor ($\gamma$) | 0.995 |
| Temperature ($\tau$) | 1.0 |
| Cluster head ($K_1$) | 10 (CIFAR-10, STL-10), 20 (CIFAR-100) |
| Cluster head ($K_2$) | 40 |
| Cluster head ($K_3$) | 160 |

Table 10: Hyperparameters used in unsupervised clustering experiments of Section 4.3

### B.2 CLUSTERING QUALITY COMPARISON

| Fwk | Method | NMI / ACC / ARI | | |
|---|---|---|---|---|
| | | CIFAR-10 | CIFAR-100 | STL-10 |
| End-to-End | K-means | 0.09 / 0.23 / 0.05 | 0.08 / 0.13 / 0.03 | 0.13 / 0.19 / 0.06 |
| | SC (Zelnik-Manor et al., 2005) | 0.10 / 0.25 / 0.09 | 0.09 / 0.14 / 0.02 | 0.10 / 0.16 / 0.05 |
| | AC (Gowda & Krishna, 1978) | 0.11 / 0.23 / 0.07 | 0.10 / 0.14 / 0.03 | 0.24 / 0.33 / 0.14 |
| | NMF (Cai et al., 2009) | 0.08 / 0.19 / 0.03 | 0.08 / 0.12 / 0.03 | 0.10 / 0.18 / 0.05 |
| | AE (Bengio et al., 2007) | 0.24 / 0.31 / 0.17 | 0.10 / 0.17 / 0.05 | 0.25 / 0.30 / 0.16 |
| | DAE (Vincent et al., 2010) | 0.25 / 0.30 / 0.16 | 0.11 / 0.15 / 0.05 | 0.22 / 0.30 / 0.15 |
| | DCGAN (Radford et al., 2016) | 0.27 / 0.32 / 0.18 | 0.12 / 0.15 / 0.05 | 0.21 / 0.30 / 0.14 |
| | DeCNN (Zeiler et al., 2010) | 0.24 / 0.28 / 0.17 | 0.09 / 0.13 / 0.04 | 0.23 / 0.30 / 0.16 |
| | VAE (Kingma & Welling, 2013) | 0.25 / 0.29 / 0.17 | 0.11 / 0.15 / 0.04 | 0.20 / 0.28 / 0.15 |
| | JULE (Yang et al., 2016) | 0.19 / 0.27 / 0.14 | 0.10 / 0.14 / 0.03 | 0.18 / 0.28 / 0.16 |
| | DEC (Xie et al., 2016) | 0.26 / 0.30 / 0.16 | 0.14 / 0.19 / 0.05 | 0.28 / 0.36 / 0.19 |
| | DAC (Chang et al., 2017) | 0.40 / 0.52 / 0.30 | 0.19 / 0.24 / 0.09 | 0.37 / 0.47 / 0.26 |
| | ADC (Haeusser et al., 2018) | - / 0.33 / - | - / 0.16 / - | - / 0.53 / - |
| | DDC (Chang et al., 2019) | 0.42 / 0.52 / 0.33 | - / - / - | 0.37 / 0.49 / 0.27 |
| | DCCM (Wu et al., 2019) | 0.50 / 0.62 / 0.41 | 0.29 / 0.33 / 0.17 | 0.38 / 0.48 / 0.26 |
| | IIC (Ji et al., 2019) | - / 0.62 / - | - / 0.26 / - | - / 0.61 / - |
| | PICA [Avg] (Huang et al., 2020) | 0.56 / 0.65 / 0.47 | 0.30 / 0.32 / 0.16 | 0.59 / 0.69 / 0.50 |
| | PICA [Best] (Huang et al., 2020) | 0.59 / 0.70 / 0.51 | 0.31 / 0.34 / 0.17 | 0.61 / 0.71 / 0.53 |
| | C2BIN [Avg] (Ours) | 0.62 / 0.72 / 0.53 | 0.36 / 0.35 / 0.20 | 0.62 / 0.73 / 0.55 |
| | C2BIN [Best] (Ours) | **0.63 / 0.73 / 0.55** | **0.38 / 0.38 / 0.22** | **0.64 / 0.75 / 0.57** |
| 2-step | SCAN (Gansbeke et al., 2020) | 0.80 / 0.88 / 0.77 | 0.49 / 0.51 / 0.33 | 0.70 / 0.81 / 0.65 |

Table 11: Full comparison with unsupervised representation models for clustering benchmark datasets. The results of previous methods are taken from Ji et al. (2019); Huang et al. (2020); Gansbeke et al. (2020).

## B.3 QUALITATIVE EXAMPLES IN IMAGE RETRIEVAL TASKS

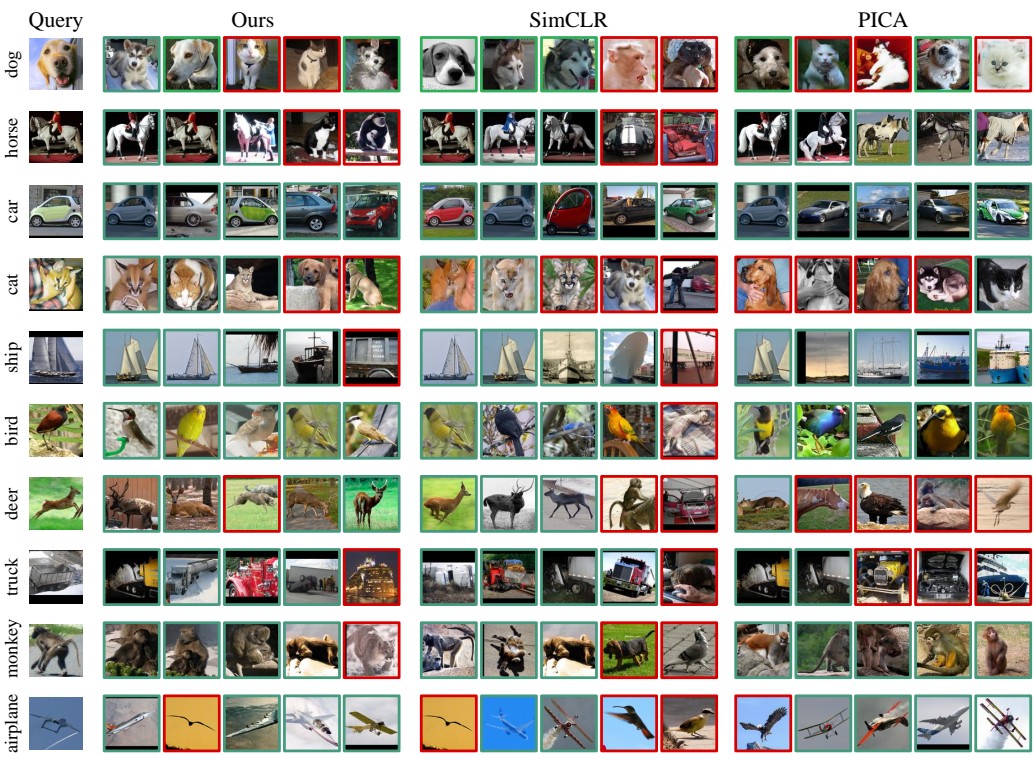

Figure 7: Top-k retrieved images by C2BIN (Columns 2-6), SimCLR (Columns 7-11), and PICA (Columns 12-17) given the query image (Column 1) from the STL-10 test set where $k$ is set as $\{1, 2, 10, 50, 100\}$.

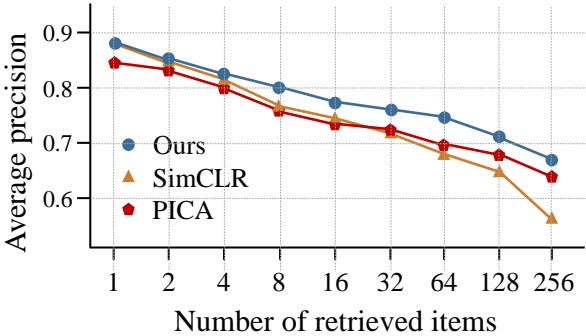

Figure 8: Image retrieval performance on STL-10 datasets.

## B.4 VISUALIZATION OF CLUSTERS

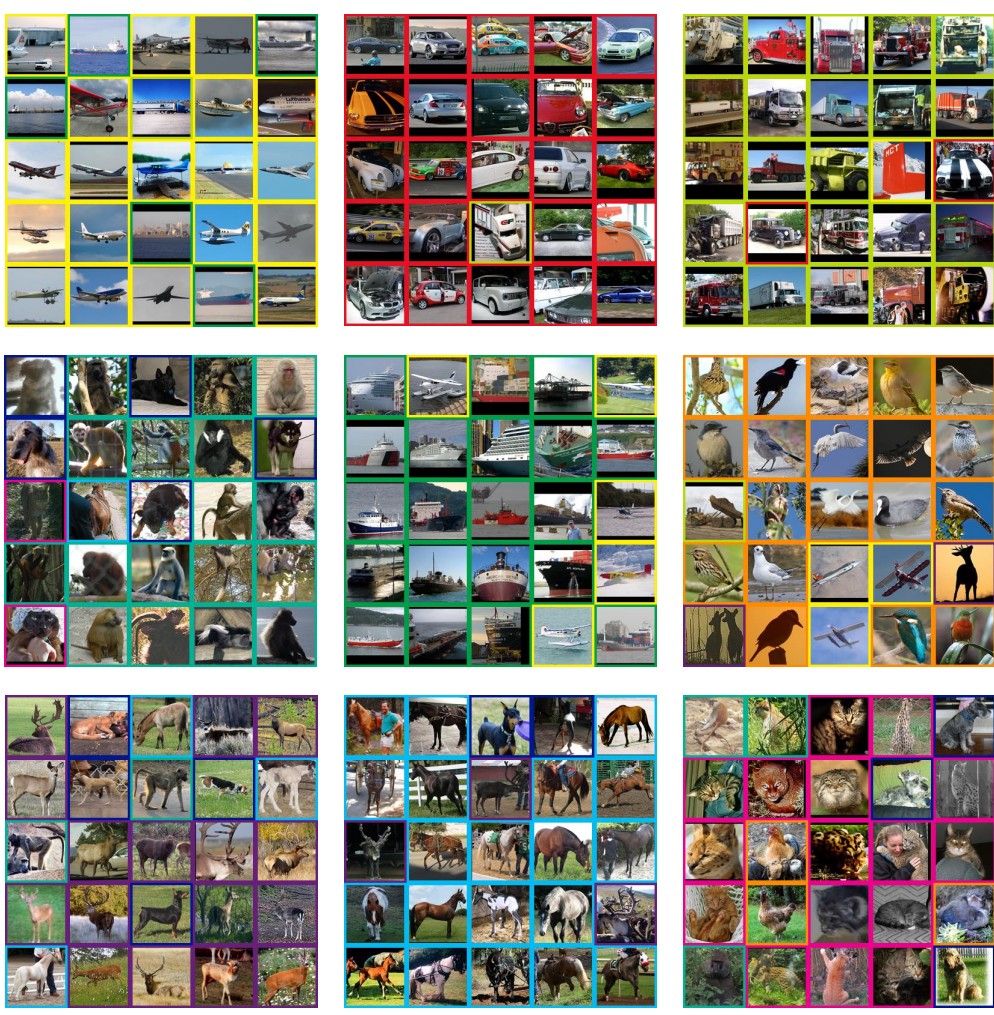

Figure 9: This figure shows a random samples of STL-10 test set images associated to the selected clusters from $k = 10$ cluster-branch. This visualization uses the experiment settings from unsupervised clustering experiment in Section 4.3. The border color enclosing each image indicates its ground-truth class.

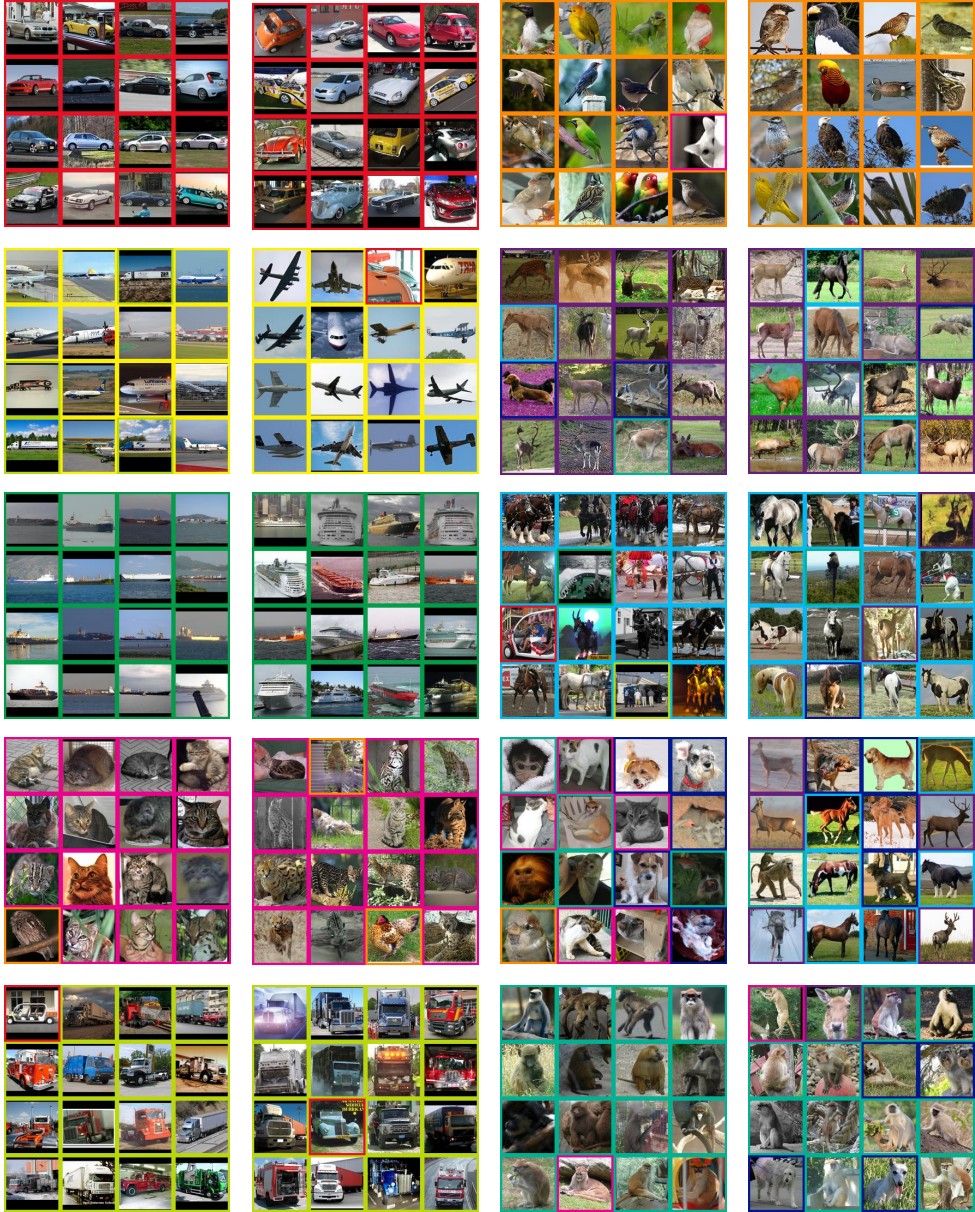

Figure 10: This figure shows a random samples of STL-10 test set images associated to the selected clusters from $k = 40$ cluster-branch. This visualization uses the experiment settings identical to Figure 9.The border color enclosing each image indicates its ground-truth class.

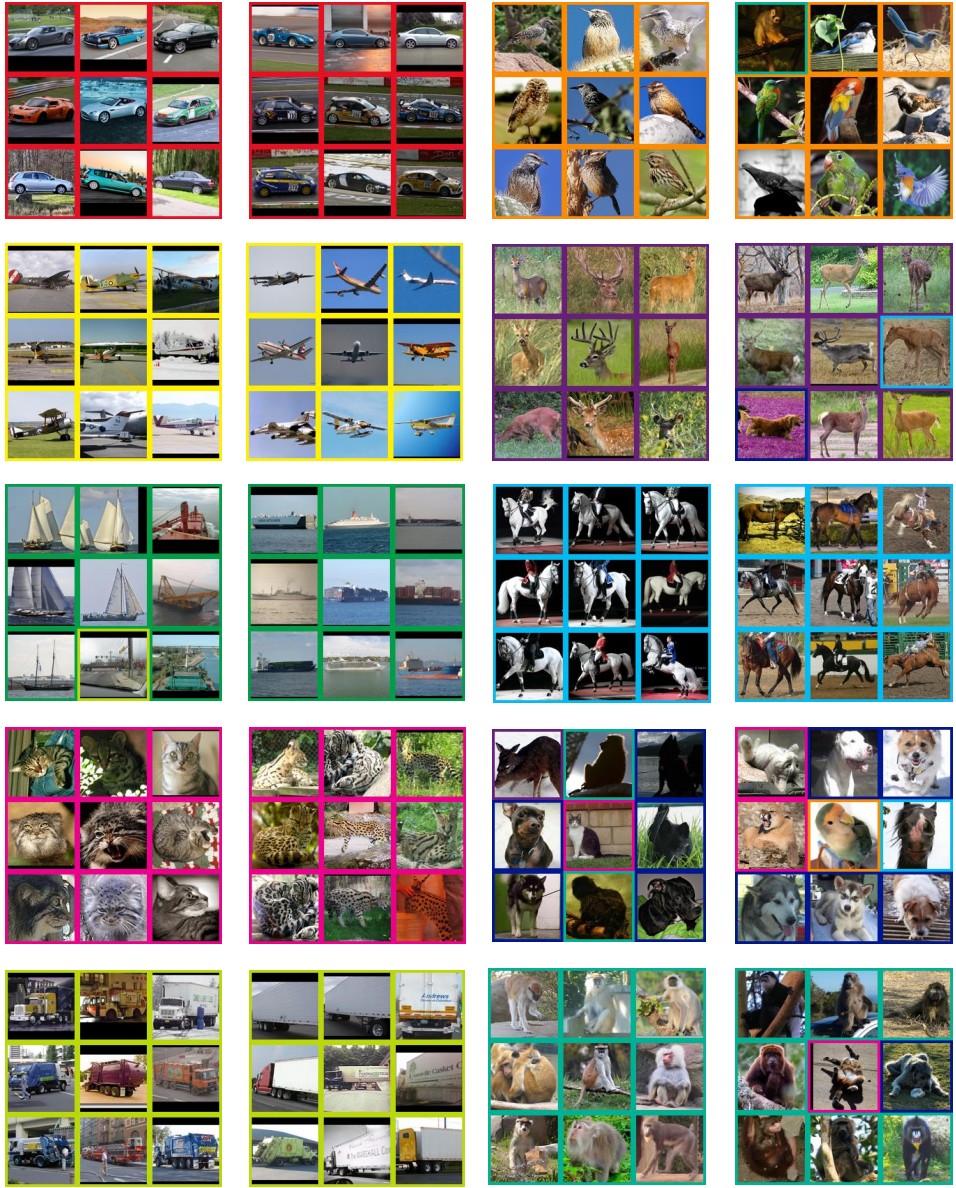

Figure 11: This figure shows a random samples of STL-10 test set images associated to the selected clusters from $k = 160$ cluster-branch. This visualization uses the experiment settings identical to Figure 9. The border color enclosing each image indicates its ground-truth class.

