# OpenReview forum: "Learning Representations by Contrasting Clusters While Bootstrapping Instances"
_ICLR.cc/2021/Conference — Reject_

### Official Review · AnonReviewer2 · 2020-10-25
**Official Blind Review #2**

**Rating:** 4
**Confidence:** 5

**Review:**

Summary:\
This paper applies batch-wise cluster assignment with bootstrapping to learn unsupervised representation. This paper claims resulting representations are better suited to non-discriminative tasks where clustering is important.
-----
+Strengths\
+This paper motivates a good direction over current unsupervised representation learning. Considering clustering performance in addition to discriminative power is a fair research question in my opinion.
+The proposed idea is interesting and seems reasonable.
+Evaluating representation learning on non-discriminative tasks is a good idea.
-----
-Concerns\
-A critical concern is the experiment setup, particularly the choice of ResNet18 as backbone and only evluating on CIFAR/STL. These datasets are quite small and are not used as the primary performance benchmark for modern unsupervised image representation learning work. This paper claims improvement in certain aspects over SimCLR, MoCo v2, and BYOL which all experiment on much larger capacity models and much larger datasets. Many insights in these prior work tie heavily to scaling to larger dataset and model capacity. This makes it difficult to compare this work.
-----
Recommendation\
The experiment setup in this paper deviates significantly from recent work of similar nature. Therefore I am not convinced by the findings presented and I recommend to reject this paper.

---

> ### Author Response · Authors · 2020-11-23
> **Answers of R2**
>
> We thank the reviewer for their clear and intuitive comments. We provide comment for their concern:
>
> In unsupervised clustering experiments, we strictly followed the guidelines from the existing literature where ResNet34 is chosen as a backbone model. We believe that the effectiveness of our approach is valid where our method has achieved state-of-the-art results among the end-to-end unsupervised clustering methods.
>
> However, we deeply understand the reviewer's concern for the potential deficiency of our method in a large-scale dataset with a high-capacity model. Therefore, we will report the experimental result for ImageNet with ResNet50 as a backbone model in the upcoming document.

---

### Official Review · AnonReviewer4 · 2020-10-27
**Good paper, interesting results on preserving cluster structures, notation is a bit confusing**

**Rating:** 6
**Confidence:** 3

**Review:**

Pros:

1- The paper presents a good solution for an important problem in self-supervised learning and contrastive learning. Proposed methods in the literature do not take the cluster structure of items into consideration. This paper proposes a hybrid loss function that aims to preserve the cluster structure (Equation 7)

2- A wide range of experiments are conducted to evaluate the proposed method. C2BIN shows a significantly better performance using the kNN classifier (particularly on CIFAR100). The good performance is also evident in clustering experiments. Thus, as the method promises the cluster structure is better preserved.

Cons:

1- The motivation part of the paper is not precise. The last sentence of the second paragraph of Section 1 states:

“However, since aforementioned instance discrimination does not consider the semantic similarities of the representations (e.g., same class), it results the learned representations to be uniformly distributed (Wang & Isola, 2020).”

It is true that semantic similarities are not considered in self-supervised and contrastive learning settings. However, this is a part of the problem as class labels do not exist. Moreover, it is not clear to me why it would lead to a uniform distribution of representations. I agree that cluster structure might be lost.

2- The notation used in Section 3 is confusing, I mention some possible misuses of notation or typos:

Section 3.1, N is defined as the number of unlabeled images. Later in Section 3.3, second paragraph, N is used as the mini-batch size.

Section 3.2 Equation (1), $P^a (z_a)$ is not properly defined. It refers to SimCLR and as I checked in the paper the same notation is not defined. Moreover, the paper should be written self-contained. Meaning that main formulation should be mentioned in the main body of the paper.

Figure 3, two siamese networks, $E_\theta$ and $E_\phi$ are not depicted in the Figure.


Equation 4, denominator, $i \neq k$, should it be $i \neq j$?

After Equation 4, it is stated that: The vectors c’ and c’’ are obtained from x’_i and x’’_ i , respectively, using the encoder $F_\theta$. This is not precise. The output of F would be r and it goes through $P^{c,k}$ to obtain z. Then c is computed using equation 3.

Comments and questions:

Table 3 shows that the choice of set K is quite important. If we are not provided with proper K and we have no access to labels, can you recommend any strategy in this case? Do you think using a fixed K with various elements is good for any dataset?

It would be nice to sort the methods of table 7 chronologically.

It is not clear how Figure 9, 10 and 11 are produced. Are the grouped images random samples and nearest neighbors?

---

> ### Author Response · Authors · 2020-11-23
> **Answers of R4**
>
> We sincerely thank the reviewer for their considerable and insightful comments. In the newly uploaded document, we marked modified parts in orange, so you can easily track the changed ones.
> Now, we provide explanations in the same order for their concerns:
>
> 1: First of all, we appreciate for pointing out the logical leap that could mislead the insights of our main motivation. Without any additional explanations, the uniformly distributed characteristics of the contrastive loss can be unclear. Recently, several works have been proposed to enhance the theoretical understandings of the contrastive loss and its variants [1,2]. Specifically, [1] has proved that, asymptotically, the contrastive loss is optimizing the alignment of positive pairs and uniformity of the induced distribution. Also, minimizing the sum of alignment loss and uniform loss exhibits the identical performance to the contrastive loss empirically. The detailed equation is stated below:
>
> $
> L_{total} = L_{align}(f:\alpha) + L_{uniform}(f:t)
> $
>
> $
> L_{align}(f:\alpha)= E_{(x,y) \sim p_{pos}}[\parallel f(x) - f(y) \parallel^{\alpha}_2]
> $
>
> $
> L_{uniform}(f:t)= \log E_{(x,y) \sim p_{data}}[e^{-t\parallel f(x) - f(y) \parallel^{2}_2}]
> $
>
> In this perspective, we believed that this uniformity constraint could push away the relevant instances and incur the representations to be not properly clustered. Therefore, our main motivation is to introduce the uniquely designed loss function, preserving the cluster structure of representations (e.g., uniform hypercube).
>
>
> 2: We deeply appreciate your meticulous comments. As R4 says, N in Section 3.3 should be replaced with n which indicates the number of instances in a mini-batch, i.e., mini-batch size.
> Please check the revised version of Section 3.3.
>
> In the Equation. (1), it is essentially motivated from [3], not [4] so we mentioned [3] at the starting point of Section 3.1 Preliminaries; however, as R4 points out, we agree that descriptions around Eq. (1) can be misleading to the reader.
> Therefore, we explicitly add [3] in front of the Equation. (1) and revised the source of $d_a$ more clearly.
> Please check the revised descriptions in Section 3.2.
>
> We correct the Equation. (4) that it means the contrastive loss among BCAs.
> Note that the number of BCAs is K because the output size of $P^{c, k}$ is K and each vector of them has $n$ dimensions because it is a batch-wise feature.
> The description of $c'$ and $c''$ are added as R4 suggested.
> Please check the revised ones in Section 3.3.
>
>
> 3: We also had the same concern as R4. Fortunately, the performance gap according to the choice of K seems marginal if our multi-scale clustering strategy (Section 3.4) is combined.
> For example, we use the same set of K for both CIFAR10 and CIFAR100 regardless of the underline labels, and C2BIN works robustly for both cases.
> We will report additional results for the choice of K in the upcoming document.
>
>
> 4: The R4's suggestion is also attractive, but we followed the convention of existing studies for Table 7.
>
>
> 5: We present the images that are randomly sampled without using the nearest neighbors.
> To be more precise, we arbitrarily select the clusters from each cluster branch $k =10, 40, 160$ for Figure 9, 10, 11 respectively. Then, images are randomly sampled for each cluster and displayed as a group.
>
> [1] Wang et al., Understanding Contrastive Representation Learning through Alignment and Uniformity on the Hypersphere.
>
> [2] Chen et al., Intriguing Properties of Contrastive Losses.
>
> [3] Grill et al, Bootstrap your own latent: A new approach to self-supervised learning
>
> [4] Chen et al, A simple framework for contrastive learning of visual representations

---

### Official Review · AnonReviewer1 · 2020-10-29
**A method combining instance-level and cluster-aware contrastive loss for representation learning**

**Rating:** 5
**Confidence:** 3

**Review:**

[Overview]

In this paper, the authors augment the instance-level self-supervised learning with cluster-aware learning mechanism during the training procedure. Specifically, for each training batch, the authors project the instances into a clustering space and then utilize a cluster-aware contrastive loss to push the augmented samples from the same instance to belong to the same cluster, otherwise for different instances. To ensure the clustering not to collapse into a single or a few cluster to find the trivial solutions, the authors further add a penalization item keep the entropy of clustering assignment be uniform to some extent. The experimental results demonstrate that the proposed method can improve the representation learning performance over SoTA methods on several datasets, while also outperforms the previous methods on clustering task. Further ablation studies show that the loss is effective to ensure the learned representation more discriminative and clusterable.

[Strength]

1. The intuition behind the proposed method is intuitive and straightforward. It has been shown in previous work like [a] that combining them together can boost the self-supervised learning performance significantly. This paper further demonstrates the promise of this direction.

[a] Unsupervised Learning of Visual Features by Contrasting Cluster Assignments. Caron et al.

2. The authors performed experiments to show that the proposed method achieves better performance on both representation learning task and clustering task, on various image datasets, such as CIFAR-10, CIFAR-100 and STL-10.

3. The ablation studies showed that the proposed clustering loss indeed helps to learn a better representation compared with the baseline model with a substantial margin, which demonstrates the effectiveness of the proposed method.

[Weakness]

1. The paper has a poor literature review of previous works. In the related work, both instance-level representation learning and deep clustering methods are not fully covered and compared. More importantly, the authors missed a very relevant and recent paper as pointed above [a]. The idea behind this paper is very similar to the above one.

2.  In this paper, the authors merely presented the results on relatively small datasets. Though it is a bit harsh to always request experiments on the large-scale dataset, such as ImageNet, proving the efficiency seems necessary especially when it is known that keep training on a large-scale dataset for a long time may dismiss the gap.

3. In table 3, it seems that only with multi-scale clustering loss, the performance will be improved across all metrics. This indicates that the proposed algorithm is a bit sensitive to the hyperparameter settings. Even with Eq.(1) + Eq.(5), the performance drops in some scenarios, which seems counterintuitive. All of these results demonstrate that the proposed method is still a bit mysterious and vulnerable.

4. The notations in the paper is hard to interpret and a bit abuse. The formula of Eq.(4) is also a bit confusing. First, what does k stands for? Second, why the denominator excludes the case of i=k if it is a regular contrastive loss.

[Summary]

Overall, I think this paper is a good trial of combining instance-level contrastive loss and deep clustering philosophy into a single learning regime, which I think is a promising direction to explore. However, as I pointed above, the novelty of the paper should be better explained. Also, according to the ablation study, the performance seems vulnerable to the choice of hyperparameters, such as cluster numbers. This increases the uncertainty about the effectiveness of the proposed method. Furthermore, the proposed method is not demonstrated on large-scale dataset such as ImageNet, which is supposed to be a routine setting on self-supervised learning community. I would recommend the authors could answer my above questions raised above.

---

> ### Author Response · Authors · 2020-11-23
> **Answers #1 of R1**
>
> We thank the reviewer for their thoughtful and detailed comments. In the newly uploaded document, we marked modified parts in orange, so you can easily track the changed ones.
> We provide comments in the same order for your concerns:
>
> 1: In our updated paper, we added the following references to the related work [1, 2, 3] and conducted the representation learning experiments from the official implementation of [1]. Although our work and [1] exploit the cluster structure to learn the image representation, there is a clear difference in high-level motivation. The goal of our work is to incorporate the cluster structure with the instance-level representation learning rather than solely relying on one of those two.
>
> Recently, methods with solely bootstrapping from the cluster structure have shown remarkable performance in the representation learning [1,4,5]. However, these results are heavily dependent on the choice of hyperparameters since pseudo-labels are noisy and ambiguous, different from the instance-level representation learning. In our experiments, [1] has only achieved 51.3% as the best accuracy in the STL-10 dataset even though we strictly follow the hyperparameter tuning guide from the authors. This sensitivity to the selection of the hyperparameters is also treated as one of the main issues in the official repository.
>
> Moreover, [1,4] utilized an optimal transport algorithm to equally partition the number of instances to the given number of clusters. However, this constraint is easily satisfied when the encoder always has the same output regardless of its input, which is the constant function. The presence of such a trivial solution also makes learning to be unstable and demanding.  On the other hand, our method learns the cluster representation through contrasting the batch-wise cluster assignments (BCAs).
> By Equation. (4), each cluster representation gradually moves toward an orthogonal direction, resulting in the confidence of the cluster assignments for each instance is softly maximized. These differences prevent our method from falling into the trivial solution (constant function) in the clustering level, bringing stability and robustness to the learning process.
>
> To support our statements, we have arranged the results of our experiments for [1] in the below table:
>
> - model: [1]
> - dataset: STL-10
> - architecture: ResNet-18
> - epochs: 200
> - batch size: 256
>
> | Prototypes(K) | LR  | Temperature | SK-epsilon  | Queue size  | Accuracy |
> |:------------------:|:----:|:-------------:|:----------------:|:------------------:|:--------------:|
> |32   | 0.3 | 0.5 | 0.03 | 0   | 54.7 |
> |128 | 1.2 | 0.5 | 0.03 | 0   | 51.1 |
> |32   | 0.3 | 0.5 | 0.03 | 768  | 49.7  |
> |128 | 0.3 | 0.5 | 0.03 | 768  | 46.2 |
> |128 | 1.2 | 0.5 | 0.03 | 768  | 44.3 |
> |128 | 0.3 | 0.5 | 0.03 | 0    | 43.8  |
> |32   | 1.2 | 0.5 | 0.03 | 768  | 41.1 |
> |128 | 4.8 | 0.5 | 0.03 | 3840 | 12.4 |
> |128 | 1.2 | 0.5 | 0.03 | 3840 | 12.2  |
> |128 | 4.8 | 0.1 | 0.05 | 3840 | 12.1 |
> |128 | 0.3 | 0.5 | 0.03 | 3840 | 11.8 |
> |128 | 0.3 | 0.5 | 0.05 | 3840 | 10.3  |
> |128 | 4.8 | 0.5 | 0.05 | 3840 | 10.2 |
> |128 | 1.2 | 0.5 | 0.05 | 3840 | 10.1 |
> |128 | 1.2 | 0.1 | 0.05 | 3840 | 10.0  |
> |128 | 0.3 | 0.1 | 0.05 | 3840 | 10.0  |
> |32   | 1.2 | 0.5 | 0.05 | 3840 | 10.0  |
> |32   | 0.3 | 0.5 | 0.05 | 3840 | 10.0  |
>
>
>
> [1] Caron et al., Unsupervised learning of visual features by contrasting cluster assignments.
>
> [2] Gansbeke et al., Scan: Learning to classify images without labels.
>
> [3] Chen et al., Intriguing Properties of Contrastive Losses.
>
> [4] Asano et al, Self-labelling via simultaneous clustering and representation learning
>
> [5] Caron et al., Deep Clustering for Unsupervised Learning of Visual Features

---

> ### Author Response · Authors · 2020-11-23
> **Answers #2, 3, 4 of R1**
>
> 2. To address the reviewer’s concern for the robustness of our method in a large-scale dataset, we will report the result for ImageNet in the upcoming document.
>
> 3. We agree with most of R1’s concern; however, we would like to recall that there is no concrete label size given in the self-supervised representation learning task.
> This can be considered as the main difference from the unsupervised image clustering studies, which directly utilize the pre-define label size in their networks.
> In our work, we found that simply combining two methods with a single and arbitrary selected cluster size (Table 3. (b),(c)) does not help to improve performances and even degenerates the representation quality.
> We guess that attaching a single cluster head after the backbone network makes its representation quality sensitive according to the head size as R1 mentioned.
> On the other hand, using multiple but single-scale cluster heads (Table 3.(d), (e)) sometimes helps the performance as shown in previous work [4] but it is marginal in our setting.
> We think the effect of multi-task learning seems small because each cluster head can capture similar patterns with others.
> Eventually, utilizing our multiple and multi-scale clustering strategy (Table 3.(f)) shows impressive results for both LP and kNN scores.
> We guess that the effect of multi-task learning is maximized because an identical representation vector should be informative enough to satisfy the following clusters, which are from abstract to detailed.
> Therefore, we think Section 4.4 and Table 3. demonstrate both the challenging point of this task and the effects of our main findings effectively.
> We will report additional results for the choice of K in the upcoming document.
>
>
> 4. We apologize for the misused typos especially for Equation. (4). Because the cluster contrastive loss is designed to calculate relative distances among the cluster features (BCAs), K indicates the number of clusters.
> The reason why the denominator excludes the case of j=i is that it is the self one.
> In our revised document, we correct misused notations and add informative descriptions for the equation.
> Please refer to Section 3.3.

---

### Public Comment · ~Wouter_Van_Gansbeke1 · 2020-11-10
**Missing relevant prior work**

Dear authors and reviewers,

As pointed out by R1, several important related works were omitted from the literature review. In particular, we would like to refer the authors to [A]. This work was accepted as a conference paper at ECCV’ 20 and holds the current state-of-the-art in unsupervised image clustering on CIFAR10, CIFAR100-20 and STL-10. Moreover, it was shown that the method from [A] could be applied to cluster large-scale datasets like ImageNet (1000 classes).

Given that the code of this paper was made publicly available, and can be found on the well-known ‘Papers With Code’ platform (https://paperswithcode.com/paper/learning-to-classify-images-without-labels), we find it unfortunate that it was omitted from the comparison. We look forward to future discussions.

[A] Van Gansbeke, Wouter, et al. "Scan: Learning to classify images without labels." ECCV. 2020.

---

### Author Response · Authors · 2020-11-25
**Final paper update**

Dear all reviewers.

We uploaded our newly updated document to answer the remained questions from the last comments.

In the new version, we added Section A.3 to discuss the impact of the choice of K.
We include exceptional cases to consider the relationship between the selected K and the underlining label size.
Please refer to Section A.3 for details.

Moreover, we attached Section 4.2 to show results on the large-scale benchmark dataset.
We found that our method trained with the same recipe of the unified setting shows performance degradation in the situation where the batch size is much smaller than the size of K; however, we expect to catch up the gap by adopting existing techniques. such as queue, as shown in the previous work.

---

### Decision · Program_Chairs · 2021-01-07
**Final Decision**

**Decision:**

Reject

**Comment:**

The idea of combining instance-level contrastive loss and deep clustering is a promising direction in recent unsupervised/self-supervised visual representation learning studies. However, authors did a poor literature review and did not cite and compare with quite a few recent popular work exploring the similar direction. The proposed methodology is not particular novel and the experimental results are also not convincing. Overall, the paper explored a promising research direction, but the paper quality is clearly below acceptance bar.